# Induced-fit expansion and contraction of a self-assembled nanocube finely responding to neutral and anionic guests

Yi-Yang Zhan[1], Tatsuo Kojima[1], Takashi Nakamura [1,4], Toshihiro Takahashi[1], Satoshi Takahashi[1], Yohei Haketa[2], Yoshiaki Shoji [3], Hiromitsu Maeda[2], Takanori Fukushima [3] & Shuichi Hiraoka [1]

Induced-fit or conformational selection is of profound significance in biological regulation. Biological receptors alter their conformation to respond to the shape and electrostatic surfaces of guest molecules. Here we report a water-soluble artificial molecular host that can sensitively respond to the size, shape, and charged state of guest molecules. The molecular host, i.e. nanocube, is an assembled structure consisting of six gear-shaped amphiphiles (GSAs). This nanocube can expand or contract its size upon the encapsulation of neutral and anionic guest molecules with a volume ranging from 74 to 535 Å$^3$ by induced-fit. The responding property of this nanocube, reminiscent of a feature of biological molecules, arises from the fact that the GSAs in the nanocubes are connected to each other only through the hydrophobic effect and very weak intermolecular interactions such as van der Waals and cation-π interactions.

[1] Department of Basic Science, Graduate School of Arts and Sciences, The University of Tokyo, 3-8-1 Komaba, Meguro-ku, Tokyo 153-8902, Japan. [2] Department of Applied Chemistry, College of Life Sciences, Ritsumeikan University, 1-1-1 Noji-higashi, Kusatsu, Shiga 525-8577, Japan. [3] Laboratory for Chemistry and Life Science, Institute of Innovative Research, Tokyo Institute of Technology, 4259 Nagatsuta, Midori-ku, Yokohama 226-8503, Japan. [4] Present address: Division of Chemistry, Faculty of Pure and Applied Sciences, University of Tsukuba, 1-1-1 Tennodai, Tsukuba, Ibaraki 305-8571, Japan. Correspondence and requests for materials should be addressed to S.H. (email: chiraoka@mail.ecc.u-tokyo.ac.jp)

Induced-fit or conformational selection is a general strategy to attain a tight binding between a molecular host and guest(s)[1–10], to induce signal transduction[11–15], and to confer allosteric regulation through conformational changes upon binding[16–23]. As often seen in biological receptors, conformational flexibility that enables molecular hosts to respond to the shape and electrostatic surface of guest molecules is a general requirement for induced-fit binding. Most of artificial molecular hosts possessing a confined binding site are rigid structures whose conformational change upon binding of guest molecule(s) is smaller than that in biological receptors. Recently, we reported a cube-shaped molecular assembly, i.e., nanocube, from six molecules of gear-shaped amphiphiles (GSAs) such as $1Cl_2$ (Fig. 1a) in water[24–26]. The structure of this nanocube is formed and maintained by only meshing between the GSAs, for which very weak, non-directional van der Waals (vdW) and cation-π interactions are responsible. Since the relative position of the GSAs is not much restricted, a variety of hydrophobic guest molecules with different molecular size and shape are expected to be accommodated in a nanospace (ca. 1 nm³) surrounded by hydrogen atoms on the aromatic rings of the GSAs. Furthermore, a polycationic character of the nanocube due to the pyridinium groups may facilitate encapsulation of charge-dispersed anionic species, whose binding must nevertheless overcome the energy required to release the water molecules around the anions[27–38].

Here we report the expansion and contraction of a water-soluble artificial molecular host that responds to the size, shape, and charged state of guest molecules with a calculated volume ranging from 74 to 535 Å³. The encapsulation of neutral molecules in the nanocube always causes the expansion. On the other hand, when anionic species are encapsulated, the nanocube shrinks responding to the negative charge of the guests. Thus, the response of the nanocube to guest molecules depends not only on the size and shape of the guests but also on their charged state.

## Results

**Structure of nanocube $1_6$.** The structure of the nanocube belongs to the $S_6$ point group. All the six GSAs in the nanocube are chemically equivalent but the symmetry of each GSA in the nanocube is reduced to be the $C_1$ point group, indicating that the three p-tolyl methyl groups (red methyl groups in Fig. 1a) of each **1** in the nanocube are chemically inequivalent (signals marked with red solid circles in Fig. 2). The structure of the nanocube is interpreted by comparing it to the Earth (Fig. 1b). One of the methyl groups in **1** ($Me^P$ in Fig. 1b) is placed around the north or the south pole, while the others ($Me^E$ in Fig. 1b) are placed near the equator. ¹H NMR signals of these p-tolyl methyl groups are observed in the upfield region due to the shielding effect caused by the neighboring aromatic rings in the nanocube[24], so the chemical shift of the p-tolyl methyl signals is a good indicator to assess the molecular meshing between the GSAs in the nanocube.

**Expansion of nanocube $1_6$.** When neutral, hydrophobic aliphatic (C3–C24) and aromatic molecules were added in an aqueous solution of $1_6Cl_{12}$, a further desymmetrization of the three p-tolyl methyl signals was not observed though the symmetry of the guest molecules is not the same as that of the nanocube, indicating the faster tumbling of the guest molecule(s) in the nanocube than the NMR timescale. (Fig. 2 and Supplementary Figs. 1–6). The aromatic region of the ¹H NMR spectra of the nanocube encapsulating the guests also changed, suggesting a slight structural change of the nanocube by induced-fit. All the signals for the guest molecules shifted to downfield by ca. 0.8 ppm, compared with those of free guest molecules in $CD_3OD$ (Supplementary Fig. 10) due to the deshielding effect caused by the

aromatic rings of the GSAs, where the phenylene groups of the propeller-shaped hexaphenylbenzene framework are nearly perpendicular to the faces of the nanocube. The smallest guest molecule encapsulated is n-propane (74 Å³), three molecules of which were cooperatively encapsulated in the nanocube, while the largest one is n-tetracosane (535 Å³) (A list of guest molecules encapsulated is shown in Supplementary Table 1).

Considering the fact that the molecular lengths of long alkanes with all anti configuration (e.g., 15.7 Å for decane) are longer than the side of the inner space of the nanocube (ca. 10 Å), long linear alkanes must be folded so as to be properly encapsulated in the cavity. However, only two kinds of ¹H NMR signals for all the alkanes (the terminal methyl groups (1.90 ppm) and all methylenes (2.33 ppm)) were observed (Supplementary Figs. 3–6), which does not tell anything about the conformation of the guest molecules in the nanocube, suggesting that even though

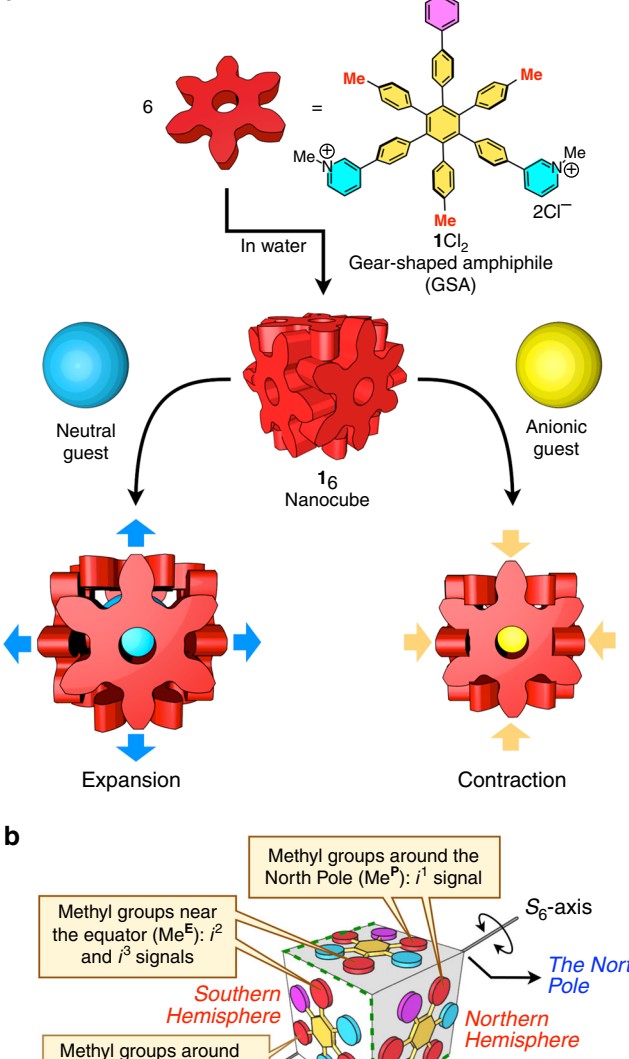

**Fig. 1** The expansion and contraction feature of the nanocube. **a** The expansion and contraction of the $1_6Cl_{12}$ nanocube assembled from six molecules of $1Cl_2$ by neutral and anionic guests, respectively. **b** Schematic representation of the structure of the nanocube. The signals $i^1$–$i^3$ indicate the chemically inequivalent p-tolyl methyl ¹H NMR signals of the nanocube in Fig. 2

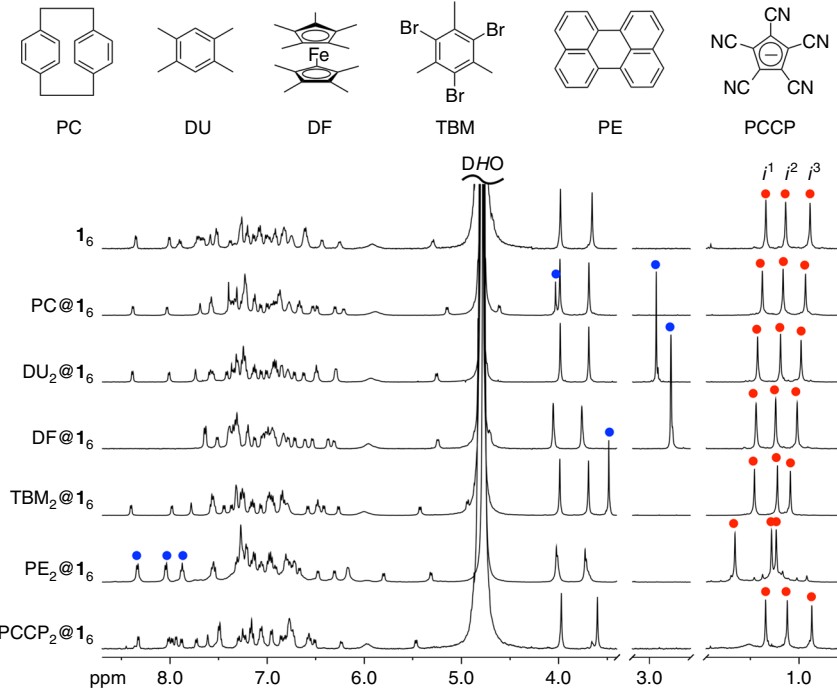

**Fig. 2** $^1$H NMR spectra of the host-guest complexation between the nanocube and guest molecules (500 MHz, D$_2$O, 298 K). Red solid circles indicate the $p$-tolyl methyl signals ($i^1$, $i^2$, and $i^3$). Blue solid circles indicate the signals for guest molecules encapsulated in the nanocube. The most downfield-shifted four signals (DF@1$_6$ and PE$_2$@1$_6$) derived from the protons neighboring nitrogen atoms of the $N$-methylpyridinium groups in 1$_6$ disappeared through the H/D exchange with D$_2$O upon heating at 90 °C[24]

long linear alkanes adopt a folded conformation, the environment of all the methylene protons are magnetically equivalent in the NMR timescale. This result is different from the previous finding that coiled alkanes are encapsulated in an artificial molecular capsule[39]. The size and shape of the nanocube cavity should allow the guest molecules to adopt a variety of different folding patterns. Aromatic molecules from one molecule of [2.2] paracyclophane (PC) (248 Å$^3$) to two molecules of perylene (PE) (508 Å$^3$) were also encapsulated in the nanocube. These results indicate the high adaptability of the nanocube to a wide range of molecular size of guests.

Induced-fit expansion of the nanocube by neutral guest molecules is confirmed by $^1$H DOSY spectroscopy. The log$D$ value of the nanocube encapsulating two molecules of 1,3,5-tribromomesitylene (TBM), –9.96, is slightly smaller than that of the empty nanocube, –9.89 (Supplementary Fig. 11), indicating the expansion of the nanocube upon the encapsulation of TBM by loosening the molecular meshing between the GSAs. The volume of the TBM$_2$@1$_6$ estimated from the hydrodynamic radius determined by $^1$H DOSY, 18 Å, is 24,000 Å$^3$, which is 1.7 times larger than that of the free nanocube (14,000 Å$^3$). Interestingly, a sum of the chemical shift changes of the three $p$-tolyl methyl signals ($i^1$–$i^3$ in Fig. 2) linearly correlates to the total volume of the guest molecules (Fig. 3a), suggesting that the nanocube expands its inner space to properly accommodate the guests. A $^1$H-$^1$H NOESY cross peak between $i^2$ and $i^3$ was observed (Supplementary Fig. 15), indicating that $i^2$ and $i^3$ are the $p$-tolyl methyl groups near the equator (Me$^E$) and that the other one ($i^1$) is around the north or the south pole (Me$^P$). In the cases where two molecules of anthracene and two molecules of perylene (PE) were encapsulated in the nanocube, large chemical shift changes of Me$^P$ ($i^1$) were observed (Fig. 3b), suggesting a large distortion around the poles by the encapsulation of a rodlike or a large planar molecule with the longest side of 12 Å. This is consistent with the previous finding that the molecular meshing

around the equator is stronger than that around the poles[40,41]. In every case, a further desymmetrization of the nanocube upon the complexation was not observed by $^1$H NMR spectroscopy, indicating that the tumbling motion of the guest molecule(s) in the nanocube is much faster than the NMR timescale.

Variable-temperature $^1$H NMR spectroscopy indicates that the thermal stability of the nanocube became higher by binding of neutral guests and depended on the volume of the guests (Table 1, Supplementary Table 2, and Supplementary Figs. 18–21). TBM stabilized the nanocube best. A larger guest than TBM, perylene (PE), stabilized the nanocube less sufficiently than TBM, indicating that guest molecules with a total volume of about 450 Å$^3$ are the best for the stabilization of the nanocube.

**Contraction of nanocube 1$_6$.** Next, the encapsulation of anionic species was investigated (Fig. 2). The titration experiment indicated that two molecules of pentacyanocyclopentadienide (PCCP) were cooperatively encapsulated in the nanocube (Supplementary Fig. 7). When NaPCCP was added in a solution of the nanocube, the $p$-tolyl methyl signals slightly shifted to upfield, even though the total volume of two molecules of PCCP (386 Å$^3$) is slightly smaller than the volume of decamethylferrocene (DF) (407 Å$^3$) (Fig. 3a and Supplementary Fig. 7). A similar upfield shift of the $p$-tolyl methyl signals was observed by the encapsulation of one molecule of CHB$_{11}$Cl$_{11}$$^-$ (CB) (403 Å$^3$) (Fig. 3a and Supplementary Fig. 8)[42]. $^1$H DOSY spectroscopy of a solution of a mixture of PCCP$_2$@1$_6$ and 1$_6$ indicated that the log$D$ value of PCCP$_2$@1$_6$, –9.85, is larger than that of 1$_6$, –9.89 (Supplementary Fig. 12). The molecular volume of PCCP$_2$@1$_6$ estimated from its hydrodynamic radius (14 Å) is 11500 Å$^3$, which is about half of that of TBM$_2$@1$_6$ (24000 Å$^3$). The longitudinal relaxation time ($T_1$) of the $p$-tolyl methyl protons of PCCP$_2$@1$_6$, 2.56 s, is longer than that of 1$_6$, 2.15 s (Supplementary Table 3), indicating that the motion of the GSAs of the PCCP$_2$@1$_6$ is restricted upon the

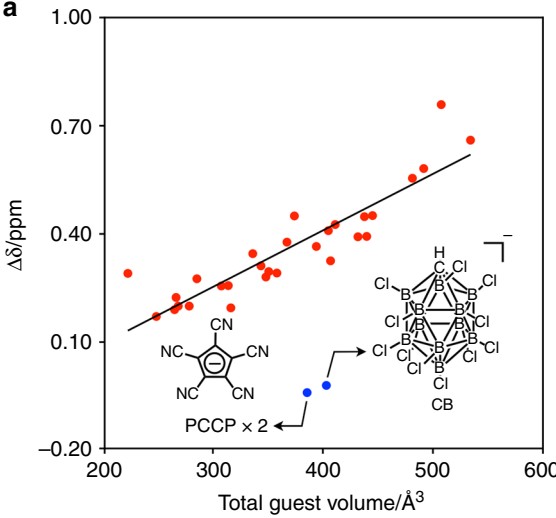

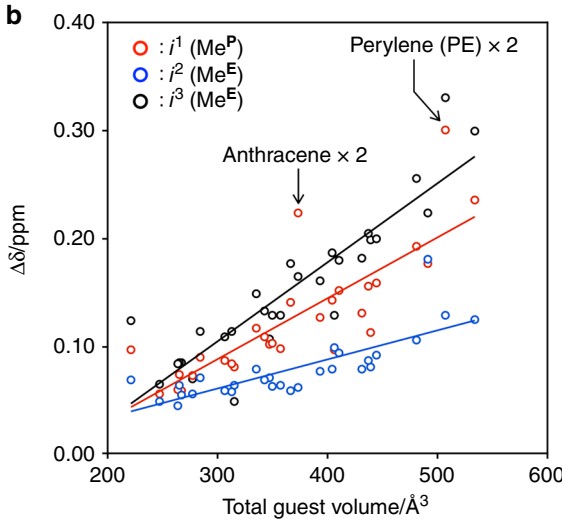

**Fig. 3** Chemical shift change of the *p*-tolyl methyl signals. **a** Plots of total chemical shift changes of the *p*-tolyl methyl signals (the sum of chemical shift changes of the three *p*-tolyl methyl signals) upon the encapsulation of guest molecules. Red and blue solid circles indicate neutral and anionic guests, respectively. **b** Plots of chemical shift change of each *p*-tolyl methyl signal ($i^1$, $i^2$, and $i^3$) upon the encapsulation of neutral guest molecules. The guest molecules tested are summarized in Supplementary Table 1

| Nanocube | Total guest volume (Å³) | $T_{1/2}$[a] (°C) | Formation ratio (6 [guest(s)@$1_6$]/[1]) at 150 °C |
|---|---|---|---|
| $1_6$ | 0 | 130 | – |
| PC@$1_6$ | 248 | 140 | – |
| DU₂@$1_6$ | 348 | 145 | – |
| DF@$1_6$ | 407 | >150[b] | 2 |
| TBM₂@$1_6$ | 440 | >150[b] | 4 |
| PE₂@$1_6$ | 508 | 143 | – |

**Table 1 Disassembly temperatures ($T_{1/2}$) for the $1_6$ nanocubes with or without guests**

[a]$T_{1/2}$ is the temperature at which half of the nanocubes are disassembled into the monomers determined by variable temperature $^1$H NMR spectroscopy ([**1**]$_{total}$ = 1.0 mM, D₂O, in a sealed pressure tube)
[b]The exact $T_{1/2}$ could not be determined because of the temperature limit of the instrument

contraction of the nanocube.[43–45] These results indicate that the nanocube shrunk upon binding of anionic guests, which is probably due to the electrostatic interactions between the nanocube with twelve positive charges and the anionic guests. As expected, cationic molecules such as tetra-*n*-butylammonium (446 Å³) were not encapsulated in the nanocube (Supplementary Fig. 9). As a consequence, the nanocube recognizes the size, shape, and charged state of guest molecules to slightly alter its structure (expansion, contraction, and/or distortion) as a response upon binding.

As NaPCCP and CsCB are soluble in water, ITC titration experiments were carried out to determine the thermodynamic parameters of the binding of the anionic guests in the nanocube (Supplementary Fig. 23), which enabled us to discuss the driving force of the encapsulation. In the case of PCCP, the binding constants for the first binding (PCCP + $1_6$ ⇌ PCCP@$1_6$) and the second binding (PCCP + PCCP@$1_6$ ⇌ PCCP₂@$1_6$) are $K_1 = 6.8 \times 10^5$ M$^{-1}$ and $K_2 = 4.1 \times 10^6$ M$^{-1}$, respectively, indicating

positive cooperativity, which is consistent with the $^1$H NMR titration experiment. As all the ionic species are well solvated, the effect of counter ions is negligible. The first binding exhibits highly negative enthalpy and entropy changes ($\Delta H_{298} = -57.5$ kcal mol$^{-1}$, $\Delta S_{298} = -166$ cal mol$^{-1}$ K$^{-1}$), which could be partly due to the chaotropic effect[46]. The introduction of electron-withdrawing groups in the cyclopentadienyl anion causes dispersion of the π electrons to lead to high polarizability of PCCP as seen in ClO₄$^-$. Upon the encapsulation of such chaotropic anions in the nanocube, the reformation of the water molecules that surrounded the anions restores hydrogen bonds between water molecules to make more ordered water network. Moreover, stronger electrostatic and vdW interactions between PCCP and the nanocube than the hydration of PCCP and the release of energetically destabilized water molecules in the cavity of the nanocube upon binding of PCCP also account for the large negative enthalpy change[47,48]. The entropic disadvantage is due to conformational fixation of the GSAs by the induced-fit contraction of the nanocube. The second binding is enthalpically disfavored ($\Delta H_{298} = 2.4$ kcal mol$^{-1}$) because of the electrostatic repulsion between the two negatively charged PCCP molecules close in vicinity in the cavity of the nanocube, so encapsulation of the second PCCP molecule was promoted by entropy ($\Delta S_{298} = 38.2$ cal mol$^{-1}$ K$^{-1}$). Electrostatic attractions in water are generally favored entropically due to desolvation of the structured water molecules around ionic species[49,50]. As the entropic loss arising from the induced-fit binding has already been paid in the first binding, the desolvation of water molecules around ions mentioned above would mainly contribute to the second binding. As to the encapsulation of CB, the binding constant is $1.46 \times 10^7$ M$^{-1}$, which is about 20 times higher than the first binding constant for PCCP. The large negative enthalpy and entropy changes ($\Delta H_{298} = -68.8$ kcal mol$^{-1}$, $\Delta S_{298} = -198$ cal mol$^{-1}$ K$^{-1}$) indicate a similar binding event to the first binding of PCCP. The stronger binding for CB than for PCCP is because larger and higher polarizable CB restores more hydrogen bonds between water molecules and makes stronger vdW interactions between CB and the nanocube upon binding.

As the nanocube shrinks upon binding of anionic guests, the anionic species and the six GSAs in the host-guest complexes more tightly bind than the neutral guests and the six GSA, higher thermal stability of the nanocube is expected by binding of anionic guest(s). Indeed, the formation ratio of PCCP₂@$1_6$ (6 [PCCP₂@$1_6$]/[1] = 60) at 130 °C is higher than that of TBM₂@$1_6$ (6[TBM₂@$1_6$]/[1] = 30) at the same temperature (Supplementary Fig. 22), indicating that the PCCP₂@$1_6$ nanocube is thermally more stable than the TBM₂@$1_6$ nanocube.

## Discussion

In conclusion, a fine structural change of the nanocube responding to the size, shape, and charged state of various guest molecules, whose volume ranges from 74 to 535 $Å^3$, has been demonstrated. Neutral guest molecules induced the expansion of the nanocube, while anionic guests the contraction of the nanocube due to electrostatic interaction between the polycationic host and the anionic guest(s) to lead to tight host-guest complexes. The nanocube can alter the size of its inner space from 14 Å (11500 $Å^3$) to 18 Å (24000 $Å^3$) depending on the size and the charged state of guest molecules. This high induced-fit property of the nanocube is because the GSAs in the nanocube are not connected by directional chemical bonds but only by molecular meshing, where vdW and cation-π interactions and the hydrophobic effect contribute to the adhesive force between the GSAs. In biological systems, not large but fine conformational changes precisely responding to the input (guest) are inevitable to transport signals to other molecules. Although vdW interactions are the weakest molecular interaction between atoms and has often been underestimated, one of the reasons why vdW interactions are well utilized in biological systems would be because vdW interactions endow assembled structures with a sensitive responding property.

## Methods

**General**. $^1H$ and other 2D NMR spectra were recorded using a Bruker AV-500 (500 MHz) spectrometer. A high pressure valved NMR tube (TCI, S-5-500-HW-HPV-7) was used for variable temperature $^1H$ NMR measurements over 100 °C in $D_2O$. All reagents were obtained from commercial suppliers (TCI Co., Ltd., WAKO Pure Chemical Industries Ltd., KANTO Chemical Co., Inc., and Sigma-Aldrich Co.) and were used as received. **1** was prepared according to the literature[24].

**Host-guest complexation between $1_6$ and guest molecules**. For the encapsulation of liquid guests not soluble in water, 1 μL of guest was added via a syringe to a $D_2O$ solution of $1_6$ ([**1**] = 1.0 mM, 600 μL) in an NMR tube. The suspension was mixed by inverting the NMR tube 4 times and sonicated for 5 min. For the encapsulation of solid guests not soluble in water, 1 mg of guest was added to a $D_2O$ solution of $1_6$ ([**1**] = 1.0 mM, 600 μL) in an NMR tube. The suspension was heated at 70 °C overnight. For large hydrophobic guest molecules, heating at 90 °C was required for the completion of the encapsulation. During heating of the suspension, the hydrogen atoms neighboring the positively charged nitrogen atoms of the N-methylpyridinium rings of **1** were exchanged with deuterium atoms[24]. For 1,3,5-triiodomesitylene, the hydrogen atoms on the N-methyl groups of **1** were also exchanged with deuterium atoms through the very long heating. For the encapsulation of gaseous molecules, guest molecules were added through bubbling in a $D_2O$ solution of $1_6$ ([**1**] = 1.0 mM, 600 μL) for 2 min. The stoichiometries between $1_6$ and the guest except the anionic guests, PCCP (pentacyanocyclopentadienide) and CB (CHB$_{11}$Cl$_{11}^-$), were determined by the integrals of the $^1H$ NMR signals of the encapsulated guest. As to PCCP and CB, the stoichiometry between $1_6$ and the encapsulated anionic guest was determined by titration experiment using the $^1H$ NMR signals for the nanocube encapsulating the anionic guest. $^1H$ NMR spectra of host-guest complexes of $1_6$ are provided in Supplementary Figs. 1–10 and details on encapsulation behavior of $1_6$ are summarized in Supplementary Table 1. $^1H$ DOSY NMR spectra of host-guest complexes of $1_6$ are provided in Supplementary Figs. 11–12.

**Determination of the positions of three p-tolyl methyl groups in $1_6$**. All the proton signals derived from $1_6$ were assigned based on $^1H$-$^1H$ COSY and $^1H$-$^1H$ NOESY spectra (Supplementary Figs. 13–15). The positions of p-tolyl methyl groups, $i^1$, $i^2$, and $i^3$ were further determined by the intermolecular NOE cross-peaks and geometry-optimized structure of $1_6$ (Supplementary Figs. 16 and 17).

**Determination of disassembly temperatures ($T_{1/2}$) of host-guest complexes of $1_6$**. Disassembly temperature ($T_{1/2}$) of the nanocubes, at which half of the nanocubes are disassembled into the monomers, was determined by variable temperature $^1H$ NMR measurements to compare the integral values of the p-tolyl methyl signals for the host-guest complexes of $1_6$ and for the monomer GSA. $^1H$ NMR spectra are provided in Supplementary Figs. 18–22 and the data are provided in Supplementary Table 2.

**Dilution ITC experiments**. Dilution isothermal titration calorimetry (ITC) experiments were conducted on a Malvern MicroCal iTC$_{200}$. Titration curves are provided in Supplementary Fig. 23.

**Relaxation measurements**. Longitudinal relaxation times ($T_1$) of the monomer GSA, **1**, and the $1_6$ and PCCP$_2$@$1_6$ nanocubes are provided in Supplementary Table 3.

## Data availability

The authors declare that all the other data supporting the findings of this study are available within the Article and its Supplementary Information files and from the corresponding author upon request.

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

## Acknowledgements

This research was supported by JSPS Grants-in-Aid for Scientific Research on Innovative Areas "Dynamical Ordering of Biomolecular Systems for Creation of Integrated Functions" (25102001 and 25102005), the Mitsubishi Foundation, and Sekisui Integrated Research.

## Author contributions

Y.-Y.Z., T.K., T.N., S.T., and S.H. conceived the project. S.H. prepared the manuscript and all the authors discussed the results and commented on the manuscript. Y.-Y.Z., T. N., and T.T carried out NMR measurements. Y.-Y.Z. carried out ITC measurements. Y. H. and H.M. synthesized NaPCCP. Y.S. and T.F. synthesized CsCB.

## Additional information

**Competing interests:** The authors declare no competing interests.

