## [Peer Review File · Nature Communications]

Reviewers' comments:

Reviewer #1 (Remarks to the Author):

Hiraoka and coworkers describe a self-assembled molecular cube which possesses an interior appropriate for the uptake of guest species. Depending on those, the size of the cube alters. The self-assembly of the host species is already of high interest. However, the most important aspect of this study is the host-guest behaviour which is thoroughly performed. The change of the size of the aggregate is well studied using DOSY NMR as an elegant method. I very much enjoyed to read this well written interesting paper on biomimetic host guest chemistry in water taking place at a huge artificial aggregate formed only by weak non-covalent interactions.

The authors may comment, if guest species also can interact with the cube from the outside.

Reviewer #2 (Remarks to the Author):

The manuscript entitled "Induced-fit expansion and contraction of a self-assembled nanocube finely responding to neutral and anionic guests" submitted by Hiraoka and co-workers describes the structural changes affecting a water soluble nanocube during the binding of various guest molecules. The water-soluble nanocube structure has been already described by the same group and it is based on a molecular assembly of six gear-shaped amphiphile molecules held together by van der Waals and cation- π interactions. The encapsulation of several hydrophobic guest molecules of different sizes and shapes in the nanocage is reported. Moreover, the inclusion of some anionic species has also been investigated (i.e. pentacyanocyclopentadienide PCCP and closo-dodecaborate CBH11Cl11-). The authors claim a response in the nanocube volume to the size (calculated volume ranging from 74 Å³ to 535 Å³), shape and charge state of the encapsulated guest molecules. While neutral guest molecules induce the expansion of the cube, anionic guests induce its contraction due to electrostatic interactions.

The major claim of the paper is the ability of the synthetic nanocube to respond to the size, shape and charge of several guest molecules. Although the outcome of the present work is exciting and could be of interest for a wide readership, the manuscript is written in a very simplified manner, which does not convince this reader about the reached conclusions. From my point of view, the authors should revise the text of the manuscript aiming at explaining in more detail the obtained results and thus to further strengthen and support the reached conclusions.

In summary, the results reported in the manuscript are worth to be published. However, in its current form they do not meet the quality level expected for a paper published in Nature Communications. I would suggest going into much more detail in the explanation of the experimental results obtained prior to publication. I also suggest the authors to address the following issues:

1) The NMR studies are poorly described. The authors based all their discussion in the small changes of the methyl protons of the host. I would suggest going into more detail on the other changes observed. For example: a) there are guests affecting the aromatic protons of the host (pyridinium protons) such as DF but other guests slightly affect these protons; b) detail the effects on the guests protons when included within the cube; c) some guests induce more desymmetrization of the nanocube; d)...

2) The authors report in some cases the inclusions of either one or two guests. I wonder how do they

know that two guests are encapsulated. For example: a) in the case of PCCP no protons are available for integration; b) in the PE, guest signals may coincide with some host signals.

3) The observation of some of the suggested complexes by mass spectrometry would support the hypothesis, especially in the cases where more than one guest is supposed to be encapsulated.

4) There are no errors for any of the data reported. Authors may want to include errors for the determined thermodynamic parameters and DOSY measurements in order to be aware of the reliability and reproducibility of the determined values.

5) The results deriving from the ITC measurements are questionable due to the bad quality of the isotherm fitting. For example, in the ITC experiment using PCCP as guest the first binding event is not well-defined (the starting of the binding event is not defined). In short, in the current form, the values obtained in the ITC measurements are not accurate. Thus, they are not adequate to quantify the driving force of the binding. In this sense, the obtained enthalpy values for the first binding p are very large, $\Delta H > -55 \text{ kcal mol}^{-1}$. Could the authors provide references for the binding of neutral or anionic guests with capsular assemblies and compare very briefly the obtained data? Are these values common in molecular recognition in water?

6) The authors studied the complexation of several neutral guests and demonstrated that they provoked the expansion of the water-soluble nanocube. However, they only studied two anionic guests to explore its contraction. I suggests studying more than two anionic species and establishing that the encapsulation of charge guests affects the size of the complexes in general.

7) In the supporting information, the ^1H pseudo 2D DOSY experiment of a ca. 1:1 mixture of 16 and PCCP2@16 (Figure S11) showed different diffusion coefficients for the proton signals at ca. 1.2 and 1 ppm for 16 and PCCP2@16, as expected. However, the signals for the protons of 16 and PCCP2@16 at ca. 1.5 ppm were overlapped and thus, the signal was composed of ca. 50% 16 and 50% PCCP2@16. The diffusion coefficient for the later signal was $1.44 \times 10^{-10} \text{ m}^2\text{s}^{-1}$ corresponding to the PCCP2@16. I would expect to obtain an average value for the diffusion coefficient between the two species. Could the authors provide a sensible explanation for this observation? I also suggest including in the supporting information the 1D fit of the DOSY decays for the proton signals used in determining the diffusion constant value.

Reviewer #3 (Remarks to the Author):

This manuscript by Zhan et al. deals with the characterization of the complexation of various organic guests by a self-assembled molecular cube. This cleverly designed host comprises six interdigitating gear-shaped amphiphiles arranged along the cube faces. Slight variations in the degree of interdigitation allow the cube to adapt its volume to the size and shape of the included guest molecules. The authors show that these geometrical adjustments can be sensitively registered by NMR spectroscopy. They characterized the binding of various neutral guest molecule, assessed the volume of the cube by using DOSY NMR spectroscopy, and obtained qualitative information about the stability of the complexes by temperature dependent NMR spectroscopy. Similar investigations were carried out with two anionic guests and in this case complex stability could also be evaluated quantitatively by isothermal titration calorimetry.

The system described in this manuscript is very original and interesting. The work has been carried out carefully and the interpretation of the authors is fully consistent with the results. This manuscript

perfectly fits the scope of Nature Communication and acceptance is therefore recommended. A few minor revisions should be considered:

- The authors write in the introduction "anion binding in water is challenging because the stabilization arising from host-guest interactions must overcome the dehydration of water molecules around the anions". While this statement is true for (inorganic) anions with a high charge density, the two anions investigated in this work are large organic charge-dispersed anions, whose dehydration is presumably not very difficult. Please rephrase accordingly.

- The authors find that binding of CB (note the error in the molecular formula of CB on page 7: CHB11Cl11 instead of CBH11Cl11) is enthalpically strongly favored but entropically opposed. This thermodynamic signature is very similar to the one found by Nau for the binding of (structurally related) dodecaborate clusters to gamma-cyclodextrin (Assaf, K. I. et al. *Angew. Chem. Int. Ed.* 2015, 54, 6852-6856). Nau attributed the interaction of these clusters to gamma-cyclodextrin to the "superchaotropic" nature of these anions, which likely also explains binding of CB to the cube described here. The authors should shortly comment on Nau's work, cite the relevant article, and put their results in context with Nau's interpretation.

- Most of the NMR spectra shown in the manuscript and the supporting information indicate the clean formation of a single complex. However, additional minor signals appear close to the p-tolyl methyl signals of the cube in some spectra (e.g. in the spectra recorded in the presence of perylene, p-xylene, triiodomesitylene). The appearance of these signals should be mentioned and their potential causes explained.

- The spectra in the supporting information moreover indicate that complex formation causes a deshielding of the guest protons. As the complexation of guest molecules by receptors with aromatic subunits usually causes guest signals to move upfield in the NMR spectra, the different effect of complex formation on the NMR spectra of the guests should be mentioned and explained.

- Why are there two signals for the free guest (mesitylene) between 2 and 2.5 ppm in spectrum g in Figure S1?

- Rebek showed that the complexation of alkanes by certain self-assembled capsules causes the guest to adopt helical conformations. These conformations have characteristic signatures in the respective NOESY NMR spectra. Do the authors observe similar effects? Rebek's work should be cited because it is related to the work described here (Rebek Jr., *J. Chem. Commun.*, 2007, 2777-2789.).

- In Figure 3a the shift of only one methyl signal is shown, which one?

- Please use equilibrium arrows in the chemical equations on page 7.

- I recommend that the manuscript should be improved with respect to language and style. Some phrases are a bit awkward and should be revised. For example: "surrounded by aromatic hydrogen atoms" (page 2, better: hydrogen atoms on the aromatic units/rings), "the symmetry of each GSA in the nanocube is desymmetrized" (page 2, better: the symmetry is reduced), "intermolecular interactions such as [...] the hydrophobic effect" (page 1, the hydrophobic effect can mediate interactions but does not represent an interaction in itself), etc.

Response to Reviewers

To reviewer 1

We are grateful for your careful review and valuable comments. Our response is as follows. Thank you again for your time in advance.

Reviewer 1's comments and our response

Comment:

Hiraoka and coworkers describe a self-assembled molecular cube which possesses an interior appropriate for the uptake of guest species. Depending on those, the size of the cube alters. The self-assembly of the host species is already of high interest. However, the most important aspect of this study is the host-guest behaviour which is thoroughly performed. The change of the size of the aggregate is well studied using DOSY NMR as an elegant method. I very much enjoyed to read this well written interesting paper on biomimetic host guest chemistry in water taking place at a huge artificial aggregate formed only by weak non-covalent interactions.

Comment 1:

(i)The authors may comment, if guest species also can interact with the cube from the outside.

Response to the reviewer

In the case of neutral guests, there is no driving force for the guests to interact with the outer surface of the nanocube. In the presence of the nanocube, the ¹H NMR signals of the guests exhibit large downfield shift (ca. 0.8 ppm) due to the strong deshielding effect, strongly suggesting the encapsulation of the guests in the nanocube. As to anionic guests (**PCCP** and **CB**), as the reviewer suggests, the anions may contact with the outer surface of the nanocube by electrostatic interactions. DOSY measurements of a mixture of the nanocube and the anionic species showed that the diffusion coefficient of the nanocube became larger than that of the free nanocube (Supplementary Figure S12). If the anionic species interact with the outer surface of the nanocube, the diffusion coefficient of the nanocube should be smaller than or as large as that of the free nanocube. This observation confirms that the anionic species were encapsulated in the nanocube.

To reviewer 2

We are grateful for your careful review and valuable comments. Our response is as follows. Thank you again for your time in advance.

Reviewer 2's comments and our response

Reviewer's Comment

The manuscript entitled “Induced-fit expansion and contraction of a self-assembled nanocube finely responding to neutral and anionic guests” submitted by Hiraoka and co-workers describes the structural changes affecting a water soluble nanocube during the binding of various guest molecules. The water-soluble nanocube structure has been already described by the same group and it is based on a molecular assembly of six gear-shaped amphiphile molecules held together by van der Waals and cation- π interactions. The encapsulation of several hydrophobic guest molecules of different sizes and shapes in the nanocage is reported. Moreover, the inclusion of some anionic species has also been investigated (i.e. pentacyanocyclopentadienide PCCP and closo-dodecaborate CBH₁₁Cl₁₁⁻). The authors claim a response in the nanocube volume to the size (calculated volume ranging from 74 Å³ to 535 Å³), shape and charge state of the encapsulated guest molecules. While neutral guest molecules induce the expansion of the cube, anionic guests induce its contraction due to electrostatic interactions. The major claim of the paper is the ability of the synthetic nanocube to respond to the size, shape and charge of several guest molecules. Although the outcome of the present work is exciting and could be of interest for a wide readership, the manuscript is written in a very simplified manner, which does not convince this reader about the reached conclusions. From my point of view, the authors should revise the text of the manuscript aiming at explaining in more detail the obtained results and thus to further strengthen and support the reached conclusions. In summary, the results reported in the manuscript are worth to be published. However, in its current form they do not meet the quality level expected for a paper published in *Nature Communications*. I would suggest going into much more detail in the explanation of the experimental results obtained prior to publication.

Comment 1:

(i) The NMR studies are poorly described. The authors based all their discussion in the small changes of the methyl protons of the host. I would suggest going into more detail on the other changes observed. For example: a) there are guests affecting the aromatic protons of the host (pyridinium protons) such as **DF** but other guests slightly affect these protons; b) detail the effects on the guests protons when included within the cube; c) some guests induce more desymmetrization of the nanocube; d)...

Response to the reviewer

As the reviewer pointed out, upon the encapsulation of guest molecules, not only the *p*-tolyl methyl signals but also the other ¹H NMR signals of the nanocube shifted; some shifted to upfield and others to downfield, which also supports the idea of the induced-fit encapsulation of guest molecules. As we succeeded in the assignment of most of the aromatic signals of the nanocube, it may be possible to discuss the structural changes in the nanocube more precisely, but since the several aromatic rings are close to each other in the nanocube, whether a certain aromatic ¹H NMR signal shifts to upfield or downfield is determined by a delicate balance of the ring currents of neighboring aromatic rings. Thus it is not easy to discuss local structural changes in the nanocube using aromatic and pyridinium methyl protons. On the other hand, the chemical shift changes of the *p*-tolyl methyl signals caused by neutral guest molecules are much simpler. As Figure 3 indicates, a

clear linear relationship between the size of the neutral guests and the chemical shift change of the *p*-tolyl methyl signals in the nanocube was observed. That is the reason why in our paper the *p*-tolyl methyl signals were used for the discussion of the induced-fit expansion and contraction of the nanocube.

The ^1H NMR signals neighboring to the nitrogen atoms in the pyridinium groups disappeared due to the H/D exchange during the encapsulation of large neutral guest molecules by heating at 90 °C in D_2O . A brief description on the disappearance of four aromatic signals is added in the revised manuscript as follows:

The caption in Figure 2

“The most downfield-shifted four signals (DF@ $\mathbf{1}_6$ and PE $_2$ @ $\mathbf{1}_6$) derived from the protons neighboring nitrogen atoms of the *N*-methylpyridinium groups in $\mathbf{1}_6$ disappeared through the H/D exchange with D_2O upon heating at 90 °C²⁴.”

As to the chemical shift change of guest molecules, the ^1H NMR spectra of TBM, PC, PE, and DU in CD_3OD was included in the Supplementary Figure 10 and we compared the chemical shift value of the free guest and the guest encapsulated in $\mathbf{1}_6$.

As the reviewer pointed out, in many cases, when guest molecules are encapsulated in artificial hosts consisting of aromatic molecules, the ^1H NMR signals of the encapsulated guest shifted to upfield because the aromatic rings of the hosts are placed toward the inner space of the hosts. On the other hand, the ^1H NMR signals of the guests encapsulated in the nanocube always shifted to downfield, which is reasonable because the inner space of the nanocube is not surrounded by the π -face but by the hydrogen atoms of aromatic rings in the gear-shaped amphiphiles. In other words, the aromatic rings that make the inner cavity of the nanocube are perpendicular to the surface of the cavity. A brief discussion about the proton signals of the encapsulated guest is added in the revised manuscript as follows:

“All the signals for the guest molecules shifted to downfield by ca. 0.8 ppm, compared with those of free guest molecules in CD_3OD (Supplementary Fig. 10) due to the deshielding effect caused by the aromatic rings of the GSAs, where the phenylene groups of the propeller-shaped hexaphenylbenzene framework are nearly perpendicular to the faces of the nanocube.

All the ^1H NMR spectra of the nanocube encapsulating guest molecules showed three sharp *p*-tolyl methyl groups in SI and the manuscript, which is the same as the free nanocube, indicating the symmetry of the nanocube was not changed upon the encapsulation. This suggests that the tumbling of the guest molecule(s) in the nanocube is faster than the NMR time-scale. A brief discussion about the symmetry of the nanocube upon the encapsulation of guest molecules is added in the revised manuscript as follows:

“a further desymmetrization of the three *p*-tolyl methyl signals was not observed though the symmetry of the guest molecules is not the same as that of the nanocube, indicating the faster tumbling of the guest molecule(s) in the nanocube than the NMR time-scale.”

Comment 2:

(ii) The authors report in some cases the inclusions of either one or two guests. I wonder how do they know that two guests are encapsulated. For example: a) in the case of PCCP no protons are available for integration; b) in the PE, guest signals may coincide with some host signals.

Response to the reviewer

In the case of neutral guests, the stoichiometry between the nanocube and the guest were unambiguously determined by the integrals of the ^1H NMR signals of the encapsulated guests. In the case of PE, naphthalene, and anthracene, as shown in Supplementary Figure 2, the signals around 8 ppm derived from the pyridinium groups disappeared through the H/D exchange with D_2O by heating. Thus, the ^1H NMR signals of the encapsulated guest that appeared around 8 ppm did not overlap with other aromatic signals of the nanocube, which enabled us to determine the stoichiometry of these guests by the integral values.

As the reviewer pointed out, since PCCP does not possess hydrogen atoms, the stoichiometry of the host-guest complex could not be determined by the integration of the ^1H NMR signals of the nanocube and the guest (PCCP). However, as mentioned above, ^1H NMR signals of the nanocube sensitively shift upon the encapsulation of guest molecules and thus it is possible to determine the stoichiometry of the host-guest complexes only using ^1H NMR signals of the nanocube. Thanks to higher binding affinity of PCCP and CB to the nanocube, the stoichiometry of the nanocube and the guest was determined by simple titration experiment using the ^1H NMR signals of the nanocube (Supplementary Figures 7 and 8). Good fittings of ITC results for PCCP and CB also support the formation of $\text{PCCP}_2@\text{nanocube}$ and $\text{CB}@\text{nanocube}$.

Comment 3:

(iii) The observation of some of the suggested complexes by mass spectrometry would support the hypothesis, especially in the cases where more than one guest is supposed to be encapsulated.

Response to the reviewer

As the reviewer pointed out, mass spectrometry should support the encapsulation of guest molecules in the nanocube. Previously, we tried to detect the mass signals of the nanocubes by conventional ESI mass spectroscopy, only to fail. After several attempts, we finally succeeded in the observation of the mass signals of the nanocubes by native mass measurement, which is known to safely detect proteins rather than conventional ESI-mass measurements. In this work, we tried to observe the signals for the host-guest complexes by native mass measurements, only to observe the signals for the nanocube. This would be due to the instability of the host-guest complexes of the nanocube under the ionization condition, which is quite reasonable because the host-guest complexes between the nanocube and the guest molecules are formed mainly by the hydrophobic effect.

Comment 4:

(iv) There are no errors for any of the data reported. Authors may want to include errors for the determined thermodynamic parameters and DOSY measurements in order to be aware of the reliability and reproducibility of the determined values.

Response to the reviewer

In SI, we have reported the error of DOSY measurements and the determined thermodynamic parameters. Additionally, the fitting curves and diffusion coefficients for each peak were included in Supplementary Figures 11 and 12.

Comment 5:

(v) The results deriving from the ITC measurements are questionable due to the bad quality of the isotherm fitting. For example, in the ITC experiment using PCCP as guest the first binding event is not well-defined (the starting of the binding event is not defined). In short, in the current form, the

values obtained in the ITC measurements are not accurate. Thus, they are not adequate to quantify the driving force of the binding. In this sense, the obtained enthalpy values for the first binding p are very large, $\Delta H > -55 \text{ kcal mol}^{-1}$. Could the authors provide references for the binding of neutral or anionic guests with capsular assemblies and compare very briefly the obtained data? Are these values common in molecular recognition in water?

Response to the reviewer

A new figure for ITC measurement using PCCP as guest is included in supplementary Figure 23, in which the fitting curve was much improved. As to the highly negative enthalpy and entropy changes for the first binding of PCCP and the binding of CB, another reviewer kindly suggested an interesting paper reported by Prof. Nau *et al.* (Assaf, K. I. *et al. Angew. Chem. Int. Ed.* **54**, 6852–6856 (2015)), where the binding of $\text{B}_{12}\text{I}_{12}^{2-}$ to γ -CD exhibits highly negative enthalpy and entropy changes ($\Delta H = -25.0 \text{ kcal mol}^{-1}$ and $\Delta S = -61.7 \text{ cal mol}^{-1} \text{ K}^{-1}$). They attributed this unusual enthalpy-entropy correlation to the chaotropic nature of the dodecaborate with high polarizability. Prof. Zhao (Awino, J. K., Gunasekara, R. W., Zhao, Y. *J. Am. Chem. Soc.* **139**, 2188–2191 (2017)) has also reported similar thermodynamic parameters in peptide binding ($\Delta H = -91.5 \text{ kcal mol}^{-1}$ and $\Delta S = -275.9 \text{ cal mol}^{-1} \text{ K}^{-1}$). An additional discussion on thermodynamic parameters for the binding of anionic guest to the nanocube based on the chaotropic effect is added in the revised manuscript as follows:

“The first binding exhibits highly negative enthalpy and entropy changes ($\Delta H_{298} = -57.5 \text{ kcal mol}^{-1}$, $\Delta S_{298} = -166 \text{ cal mol}^{-1} \text{ K}^{-1}$), which is partly due to the chaotropic effect⁴⁶. The introduction of electron withdrawing groups in the cyclopentadienyl anion causes dispersion of the π electrons to lead to high polarizability of PCCP as seen in ClO_4^- . Upon the encapsulation of such chaotropic anions in the nanocube, the reformation of the water molecules that surrounded the anions restores hydrogen bonds between water molecules to make more ordered water network.”

Comment 6:

(vi) The authors studied the complexation of several neutral guests and demonstrated that they provoked the expansion of the water-soluble nanocube. However, they only studied two anionic guests to explore its contraction. I suggest studying more than two anionic species and establishing that the encapsulation of charge guests affects the size of the complexes in general.

Response to the reviewer

As the reviewer pointed out, it is interesting to extend the number of anionic guests to explore the contraction of the nanocube. However, we focused on the general encapsulation property of the nanocube in this paper. We employed PCCP and CB because their sizes are similar with those of the neutral guests in this study. The shrinking behavior of the nanocube upon the encapsulation of PCCP or CB clearly highlights the charge-response property of the nanocube. We would like to report the results with various anionic guests in a future work.

Comment 7:

(vii) In the supporting information, the ^1H pseudo 2D DOSY experiment of a ca. 1:1 mixture of $\mathbf{1}_6$ and $\text{PCCP}_2@1_6$ (Figure S11) showed different diffusion coefficients for the proton signals at ca. 1.2 and 1 ppm for $\mathbf{1}_6$ and $\text{PCCP}_2@1_6$, as expected. However, the signals for the protons of $\mathbf{1}_6$ and $\text{PCCP}_2@1_6$ at ca. 1.5 ppm were overlapped and thus, the signal was composed of ca. 50% $\mathbf{1}_6$ and 50% $\text{PCCP}_2@1_6$. The diffusion coefficient for the later signal was $1.44 \times 10^{-10} \text{ m}^2 \text{ s}^{-1}$ corresponding to

the PCCP₂@**1**₆. I would expect to obtain an average value for the diffusion coefficient between the two species. Could the authors provide a sensible explanation for this observation? I also suggest including in the supporting information the 1D fit of the DOSY decays for the proton signals used in determining the diffusion constant value.

Response to the reviewer

According to the reviewer's suggestion, the fitting curves of DOSY measurements are shown in Supplementary Figures 11 and 12. The 1D DOSY spectra of the mixture of PCCP₂@**1**₆ and **1**₆ were replaced with new spectra, in which the overlapped signal for PCCP₂@**1**₆ and **1**₆ exhibited different diffusion coefficient. The diffusion coefficients of PCCP₂@**1**₆ and **1**₆ were determined by the average *D* value of the two upfield signals of the *p*-tolyl methyl groups of PCCP₂@**1**₆ (1.117 ppm, 0.864 ppm) and **1**₆ (1.137 ppm, 0.887 ppm), respectively. As to the diffusion coefficient of the ¹H NMR signals at 1.339 ppm, which composed of 65% PCCP₂@**1**₆ and 35% **1**₆ based on the integral value of the *p*-tolyl methyl groups in PCCP₂@**1**₆ and **1**₆, the exact diffusion coefficient is $1.35 \times 10^{-10} \text{ m}^2 \text{ s}^{-1}$, which is between those of PCCP₂@**1**₆ ($1.40 \times 10^{-10} \text{ m}^2 \text{ s}^{-1}$) and **1**₆ ($1.28 \times 10^{-10} \text{ m}^2 \text{ s}^{-1}$). A simple description of the DOSY spectra for the mixture of PCCP₂@**1**₆ and **1**₆ is added in the revised SI as follows:

“The diffusion coefficients of PCCP₂@**1**₆ and **1**₆ were determined by the average *D* value of the two upfield signals of the *p*-tolyl methyl groups of PCCP₂@**1**₆ (1.117 ppm, 0.864 ppm) and **1**₆ (1.137 ppm, 0.887 ppm), respectively. The ¹H NMR signal at 1.339 ppm is derived from 65% of PCCP₂@**1**₆ and 35% of **1**₆ based on the integral values the *p*-tolyl methyl groups in PCCP₂@**1**₆ and **1**₆. The diffusion coefficient of the signal is $1.35 \times 10^{-10} \text{ m}^2 \text{ s}^{-1}$, which is between those of PCCP₂@**1**₆ and **1**₆.”

To reviewer 3

We are grateful for your careful review and valuable comments. Our response is as follows. Thank you again for your time in advance.

Reviewer 3's comments and our response

Reviewer's Comment

This manuscript by Zhan et al. deals with the characterization of the complexation of various organic guests by a self-assembled molecular cube. This cleverly designed host comprises six interdigitating gear-shaped amphiphiles arranged along the cube faces. Slight variations in the degree of interdigitation allow the cube to adapt its volume to the size and shape of the included guest molecules. The authors show that these geometrical adjustments can be sensitively registered by NMR spectroscopy. They characterized the binding of various neutral guest molecule, assessed the volume of the cube by using DOSY NMR spectroscopy, and obtained qualitative information about the stability of the complexes by temperature dependent NMR spectroscopy. Similar investigations were carried out with two anionic guests and in this case complex stability could also be evaluated quantitatively by isothermal titration calorimetry. The system described in this manuscript is very original and interesting. The work has been carried out carefully and the interpretation of the authors is fully consistent with the results. This manuscript perfectly fits the scope of *Nature Communication* and acceptance is therefore recommended.

Comment 1:

The authors write in the introduction "anion binding in water is challenging because the stabilization arising from host-guest interactions must overcome the dehydration of water molecules around the anions". While this statement is true for (inorganic) anions with a high charge density, the two anions investigated in this work are large organic charge-dispersed anions, whose dehydration is presumably not very difficult. Please rephrase accordingly.

Response to the reviewer

As the reviewer pointed out, the dehydration of PCCP and CB is less difficult than that of inorganic anions. However, it is still true that there is some energetic penalty of dehydration of PCCP and CB upon the encapsulation. The introduction in the manuscript was revised according to the reviewer's comments as follows:

"a polycationic character of the nanocube due to the pyridinium groups may facilitate encapsulation of charge-dispersed anionic species, though anion binding in water is challenging because the stabilization arising from host-guest interactions must overcome the dehydration of water molecules around the anions"

Comment 2:

The authors find that binding of CB (note the error in the molecular formula of CB on page 7: $\text{CHB}_{11}\text{Cl}_{11}$ instead of $\text{CBH}_{11}\text{Cl}_{11}$) is enthalpically strongly favored but entropically opposed. This thermodynamic signature is very similar to the one found by Nau for the binding of (structurally related) dodecaborate clusters to gamma-cyclodextrin (Assaf, K. I. et al. *Angew. Chem. Int. Ed.* **2015**, *54*, 6852–6856). Nau attributed the interaction of these clusters to gamma-cyclodextrin to the "superchaotropic" nature of these anions, which likely also explains binding of CB to the cube

described here. The authors should shortly comment on Nau's work, cite the relevant article, and put their results in context with Nau's interpretation.

Response to the reviewer

We deeply appreciate the reviewer's kind suggestion of the nice work reported by Prof. Nau *et al.*. The trend of the thermodynamic parameters of the binding of CB and the first binding of PCCP is similar to the results shown in this paper. An additional discussion on the thermodynamic parameters for the binding of the anionic guests to the nanocube is added in the revised manuscript as follows:

“The first binding exhibits highly negative enthalpy and entropy changes ($\Delta H_{298} = -57.5 \text{ kcal mol}^{-1}$ and $\Delta S_{298} = -166 \text{ cal mol}^{-1} \text{ K}^{-1}$), which is partly due to the chaotropic effect⁴⁶. The introduction of electron withdrawing groups in the cyclopentadienyl anion causes dispersion of the π electrons to lead to high polarizability of PCCP as seen in ClO_4^- . Upon the encapsulation of such chaotropic anions in the nanocube, the reformation of the water molecules that surrounded the anions restores hydrogen bonds between water molecules to make more ordered water network.”

Comment 3:

Most of the NMR spectra shown in the manuscript and the supporting information indicate the clean formation of a single complex. However, additional minor signals appear close to the *p*-tolyl methyl signals of the cube in some spectra (e.g. in the spectra recorded in the presence of perylene, *p*-xylene, triiodomesitylene). The appearance of these signals should be mentioned and their potential causes explained.

Response to the reviewer

The additional minor signals appear close to the *p*-tolyl methyl signals of the nanocube in some spectra were assigned in Supplementary Figures 1 and 2. The minor signals are derived from a trace amount of **1₆** encapsulating a different number of guest molecules. In the case of perylene and 1,3,5-triiodomesitylene, according to time-dependent ¹H NMR measurements, initially the intensities of these minor signals were higher and then decreased with a concomitant increase of the major signals.

Comment 4:

The spectra in the supporting information moreover indicate that complex formation causes a deshielding of the guest protons. As the complexation of guest molecules by receptors with aromatic subunits usually causes guest signals to move upfield in the NMR spectra, the different effect of complex formation on the NMR spectra of the guests should be mentioned and explained.

Response to the reviewer

As the reviewer pointed out, in many cases, when guest molecules are encapsulated in artificial hosts consisting of aromatic molecules, the ¹H NMR signals of the encapsulated guest shifted to upfield because the aromatic rings of the hosts are placed toward the inner space of the hosts. On the other hand, the ¹H NMR signals of the guests encapsulated in the nanocube always shifted to downfield, which is reasonable because the inner space of the nanocube is not surrounded by the π -face but by the hydrogen atoms of aromatic rings in the gear-shaped amphiphiles. In other words, the aromatic rings that make the inner cavity of the nanocube are perpendicular to the surface of the cavity. A brief discussion about the proton signals of the encapsulated guest is added in the revised manuscript as follows:

“All the signals for guest molecules shifted downfield by ca. 0.8 ppm, compared with those of free guest molecules in CD₃OD (Supplementary Fig. 10) due to the deshielding effect caused by the aromatic rings of the GSAs, where the phenylene groups in the propeller-shaped hexaphenylbenzene framework are nearly perpendicular to the faces of the nanocube.”

Comment 5:

Why are there two signals for the free guest (mesitylene) between 2 and 2.5 ppm in spectrum g in Figure S1?

Response to the reviewer

The observation of two kinds of signals for the free guest is due to the mesitylenes dissolved and undissolved in water, respectively. Similar result was also observed in the case of *m*-xylene. A brief description on these signals is added in the caption of Supplementary Figure 1 as follows:

“Two signals were observed for liquid free guest, mesitylene and *m*-xylene, which are derived from the free guest molecules dissolved and undissolved in water.”

Comment 6:

Rebek showed that the complexation of alkanes by certain self-assembled capsules causes the guest to adopt helical conformations. These conformations have characteristic signatures in the respective NOESY NMR spectra. Do the authors observe similar effects? Rebek's work should be cited because it is related to the work described here (Rebek Jr., *J. Chem. Commun.*, **2007**, 2777–2789.).

Response to the reviewer

As the reviewer pointed out, when we found the encapsulation of long alkanes in the nanocube, we also expected that the encapsulated alkanes would fold into a coiled conformation. However, we did not observe the splitting or characteristic signals of the coiled alkanes (C3 to C24) in the nanocube but only two kinds of ¹H NMR signals for all the alkanes (the terminal methyl (1.90 ppm) and all methylenes protons (2.33 ppm)). Considering that the side of the inner space is much shorter than the length of long alkanes with an extended conformation, such long alkanes must be folded in the nanocube. The simple ¹H NMR pattern for the encapsulated guest molecules indicates that the folded alkanes change their conformations faster than the NMR time-scale. This result different from the Rebek's report is due to the high induced-fit nature of the nanocube. Thus, as the reviewer suggested, this discussion is worth mentioning in the manuscript. We discuss this in the revised manuscript as follows, citing the Prof. Rebek's paper.

“This result is different from the previous finding that coiled alkanes are encapsulated in an artificial molecular capsule³⁹. The high induced-fit property of the nanocube would allow the guest molecules to easily change their conformation in the confined space.”

Comment 7:

In Figure 3a the shift of only one methyl signal is shown, which one?

Response to the reviewer

The plots in Figure 3a represent the total chemical shift changes of the *p*-tolyl methyl signals, that is, the sum of the chemical shift changes of the three *p*-tolyl methyl signals. To avoid the ambiguity, we have made this point clear in the caption of Figure 3 in the revised manuscript.

Comment 8:

Please use equilibrium arrows in the chemical equations on page 7.

Response to the reviewer

According to the reviewer's advice, we have changed the arrows in the chemical equations into equilibrium arrows.

Comment 9:

I recommend that the manuscript should be improved with respect to language and style. Some phrases are a bit awkward and should be revised. For example: "surrounded by aromatic hydrogen atoms" (page 2, better: hydrogen atoms on the aromatic units/rings), "the symmetry of each GSA in the nanocube is desymmetrized" (page 2, better: the symmetry is reduced), "intermolecular interactions such as [...] the hydrophobic effect" (page 1, the hydrophobic effect can mediate interactions but does not represent an interaction in itself), etc.

Response to the reviewer

The manuscript was revised according to the reviewer's advice and we have carefully checked the whole manuscript with respect to language and style.

Reviewers' comments:

Reviewer #3 (Remarks to the Author):

The authors adequately responded to my previous comments. I therefore request no further scientific revision but propose the following linguistic improvements:

Page 1, line 6: "reminiscent of a feature of biological molecules" instead of "reminiscent of a behavior of biological molecules"

Page 1, line 7: "through solvophobic and weak intermolecular interactions" instead of "through the hydrophobic effect and very weak intermolecular interactions"

Page 2, line 7: "whose binding must nevertheless overcome the energy required to release the water molecules around the anions" instead of "though anion binding in water is challenging [...] around the anions"

Page 4, line 11: "The size and shape of the nanocube cavity should allow the guest molecules to adopt a variety of different folding patterns" instead of "The high induced-fit property [...] in the confined space"

Page 7, line 25: "cooperativity" because of "allostericity" (Allosteric effects are usually associated with two different binding sites, one for the effector and one for the actual substrate. In this case, both guest are bound in the same cavity but binding of the first seems to cooperatively enhance binding of the second.)

Page 7, line 28: "which could be partly due" instead of "which is partly due"

Page 8, line 1: "The entropic disadvantage" instead of "The entropic unfavorability"

Page 8, line 5: "so encapsulation of the second PCCP molecule was promoted by entropy" instead of "so the second PCCP was encapsulated entropically"

Page 9, line 9: "VdW, cation-pi, and solvophobic interactions" instead of "vdW and cation-pi interactions and the hydrophobic effect"

Reviewer #4 (Remarks to the Author):

I have reviewed the revised version of the manuscript from Hiraoka et al., in which the authors have addressed the comments of three referees to the first version of the manuscript.

The manuscript from Hiraoka et al. describes structural changes affecting to the encapsulation of various neutral and anionic guests in a water soluble nanocube. The nanocage is assembled from six gear-shaped amphiphile molecules held together by van der Waals and cation-pi interactions. An array of structurally diverse anionic and neutral guests have been encapsulated.

Many of the concerns made by the previous referees have been addressed. However, one of them dealt with a simplified discussion of the manuscript. More detailed explanations to strengthen and support the reached conclusions were requested. In particular, reviewer 2 noted that NMR studies

were poorly described. In my opinion, not much has been done in this regard. The authors admit that there are difficulties in discussing all local structural changes in the nanocube due to encapsulation of such an array of structurally diverse guests with overall non-trivial structures. This would not be a problem if the authors would present other evidences pointing to the formation of the proposed guest@1₆. Unfortunately, m/z values in agreement with the guest@1₆ complexes in mass spectrometry measurements have not been observed/provided. Neither have crystals for X-Ray analysis been produced, not even for one complex!

The cage is not assembled by strong interactions, and I wonder how the authors can exclude that one of the faces of the cube is not released during encapsulation of the guest, especially for the large guests. For instance, how can the authors exclude that the structure of the complex corresponding to the encapsulation of decane is not decane@1₅ instead of decane@1₆. Would this alternative structure for the complex not be in agreement with a more dynamic complex with less differentiated protons in NMR, as observed (just one signal for the CH₃ and another for the CH₂ protons)?

Thus, my main criticism to this work is that the whole structure elucidation of the encapsulated complexes is weak as it is mainly based on changes of spectral data in NMR after encapsulation and in DOSY measurements (which I doubt that can unequivocally distinguish between guest@1₆ versus 1₆ or versus guest@1₅).

After careful analysis of the manuscript, I still agree with reviewer 2 in the fact that the results reported in the manuscript are worth to be published, but, in its current form they do not meet the quality expected for publications in Nature Communications.

Response to Reviewers

To reviewer 3

We are grateful for your careful review and valuable comments. Our response is as follows. Thank you again for your time in advance.

Reviewer 3's comments and our response

Comment:

The authors adequately responded to my previous comments. I therefore request no further scientific revision but propose the following linguistic improvements:

Page 1, line 6: "reminiscent of a feature of biological molecules" instead of "reminiscent of a behavior of biological molecules"

Page 1, line 7: "through solvophobic and weak intermolecular interactions" instead of "through the hydrophobic effect and very weak intermolecular interactions"

Page 2, line 7: "whose binding must nevertheless overcome the energy required to release the water molecules around the anions" instead of "though anion binding in water is challenging [...] around the anions"

Page 4, line 11: "The size and shape of the nanocube cavity should allow the guest molecules to adopt a variety of different folding patterns" instead of "The high induced-fit property [...] in the confined space"

Page 7, line 25: "cooperativity" because of "allostericity" (Allosteric effects are usually associated with two different binding sites, one for the effector and one for the actual substrate. In this case, both guest are bound in the same cavity but binding of the first seems to cooperatively enhance binding of the second.)

Page 7, line 28: "which could be partly due" instead of "which is partly due"

Page 8, line 1: "The entropic disadvantage" instead of "The entropic unfavorability"

Page 8, line 5: "so encapsulation of the second PCCP molecule was promoted by entropy" instead of "so the second PCCP was encapsulated entropically"

Page 9, line 9: "VdW, cation- π , and solvophobic interactions" instead of "vdW and cation- π interactions and the hydrophobic effect"

Response to the reviewer

Thank you for the reviewer's kind editing of our manuscript. We revised where the reviewer pointed out as the reviewer suggested except for the two points concerning solvophobic interaction. Interactions working between molecules assembled under solvophobic circumstances are mainly van der Waals (dispersion) interaction and the driving force of the assembly of molecules under these conditions is not the attractive interactions between the components but the stabilization of hydrating water molecules that are released in the bulk solvent upon the self-assembly. In other words, there is no special intermolecular interaction that can be called hydrophobic (solvophobic) interactions. This point was also emphasized by a recent review about aqueous supramolecular chemistry (*Nat. Chem.* **2018**, *10*, 8–16). Thus, we use "the hydrophobic effect" instead of "hydrophobic (solvophobic) interactions" in our manuscript.

To reviewer 4

We are grateful for your careful review and valuable comments. Our response is as follows. Thank you again for your time in advance.

Reviewer 4's comments and our response

Reviewer's Comment

I have reviewed the revised version of the manuscript from Hiraoka et al., in which the authors have addressed the comments of three referees to the first version of the manuscript. The manuscript from Hiraoka et al. describes structural changes affecting to the encapsulation of various neutral and anionic guests in a water soluble nanocube. The nanocage is assembled from six gear-shaped amphiphile molecules held together by van der Waals and cation- π interactions. An array of structurally diverse anionic and neutral guests have been encapsulated.

Many of the concerns made by the previous referees have been addressed. However, one of them dealt with a simplified discussion of the manuscript. More detailed explanations to strengthen and support the reached conclusions were requested. In particular, *1 reviewer 2 noted that NMR studies were poorly described. In my opinion, not much has been done in this regard. The authors admit that there are difficulties in discussing all local structural changes in the nanocube due to encapsulation of such an array of structurally diverse guests with overall non-trivial structures. This would not be a problem if the authors would present other evidences pointing to the formation of the proposed guest@1₆. Unfortunately, 2 m/z values in agreement with the guest@1₆ complexes in mass spectrometry measurements have not been observed/provided. Neither have crystals for X-Ray analysis been produced, not even for one complex!*

The cage is not assembled by strong interactions, and *3 I wonder how the authors can exclude that one of the faces of the cube is not released during encapsulation of the guest, especially for the large guests. For instance, how can the authors exclude that the structure of the complex corresponding to the encapsulation of decane is not decane@1₅ instead of decane@1₆.* Would this alternative structure for the complex not be in agreement with a more dynamic complex with less differentiated protons in NMR, as observed (just one signal for the CH₃ and another for the CH₂ protons)?

Thus, my main criticism to this work is that the whole structure elucidation of the encapsulated complexes is weak as it is mainly based on changes of spectral data in NMR after encapsulation and in DOSY measurements (which I doubt that can unequivocally distinguish between guest@1₆ versus 1₆ or versus guest@1₅).

After careful analysis of the manuscript, I still agree with reviewer 2 in the fact that the results reported in the manuscript are worth to be published, but, in its current form they do not meet the quality expected for publications in *Nature Communications*.

Response to the reviewer

Our response to Comment 1:

What the reviewer 2 requested is a more detailed discussion on *the solution structure* of the nanocube upon the encapsulation of guest molecules from ¹H NMR spectra, because in most artificial molecular hosts, spectral changes in the host signals are simpler than those observed in the nanocube. This dramatic change in the ¹H NMR signals arises from the high adaptability of the nanocube that sensitively responds to the size and shape of guest molecules, so the change in *the solution structure* of the nanocube by the encapsulation of guest molecules is quite interesting. Unfortunately, mass measurement (gas state) and single crystal X-ray analysis (solid state)

proposed by the reviewer 4 cannot help the discussion on very small structural change in *the solution structure* of the nanocube. Thus, we discussed the change in the shape of the nanocube by the three chemically inequivalent *p*-tolyl methyl signals (Me^{P} (i^1) signal and two kinds of Me^{E} signals (i^2 and i^3)) and found that when rodlike or planar molecules are encapsulated, large structural changes around the poles are induced, which is demonstrated by the large chemical shift change in Me^{P} signal (i^1). It is true that the analysis of the chemical shift change in a total of 42 chemically inequivalent signals of the nanocube would enable us to discuss the local structural change of the nanocube in more detail. However, even if the chemical shift changes in all the 42 signals can be followed clearly, it is not necessarily possible to obtain more detailed information about the solution structure of the nanocube than that obtained from the three *p*-tolyl groups, because the three methyl groups are widely distributed in structurally characteristic positions (the north and south poles and equator). Thus, the three methyl signals, which do not overlap with other signals, are quite useful for the discussion on the solution structure of the nanocube.

Our response to Comment 2:

As the reviewer pointed out, we have understood the importance of the direct evidence for the structure of the host-guest complexes and the stoichiometry between the nanocube and guest molecules by mass and/or X-ray analyses. Unfortunately, due to the lability of the nanocube and its host-guest complexes in gas state, where no stabilization by the hydrophobic effect can work, we could not detect mass signals for the host-guest complexes. As to X-ray analysis, we previously reported a single crystal structure of a host-guest complex between TBM and nanocube in which a benzene ring attached to the periphery of a hexaphenylbenzene core of **1** is replaced with a 3-pyridyl group and found that two molecules of TBM (440 \AA^3) are encapsulated in the nanocube (*JACS*, **2010**, *132*, 13223–13225). As the structure of the GSA **1** used in this paper is very similar to that in the crystal structure, it is quite reasonable that **1**₆ encapsulates two molecules of TBM as well, which is consistent with the stoichiometry between **1**₆ and TBM determined by ¹H NMR spectroscopy.

Our response to Comment 3:

As to the possibility of an open cube (**1**₅) with a large guest molecule (G) such as decane, we considered the symmetry of this complex, G@**1**₅. As the nanocube **1**₆ has an *S*₆ axis and a center of symmetry, all six GSAs in **1**₆ with the *C*₁ symmetry are chemically equivalent, which is confirmed by the result that three *p*-tolyl methyl signals with the same integrals for *C*₁-symmetrized GSAs were observed by ¹H NMR spectroscopy. When one of the GSAs in **1**₆ is removed, an open cube, **1**₅, loses the *S*₆ axis and the center of symmetry and the five GSAs in **1**₅ are not chemically equivalent. In particular, the environment of the methyl groups that shared the sides with the GSA removed from **1**₆ (two of the six Me^{P} groups (i^1), two of the twelve Me^{E} groups (i^2 and i^3)) should strongly be altered in **1**₅, so these methyl groups should be chemically inequivalent with the original *p*-tolyl methyl groups (Me^{P} and Me^{E} in **1**₆) (a figure shown below). According to this consideration, at least a total of six *p*-tolyl methyl signals, three Me^{P} (i^1), two Me^{P} (i^1), four Me^{E} (i^2), one Me^{E} (i^2), four Me^{E} (i^3), and one Me^{E} (i^3), would be observed in the ¹H NMR spectrum of G@**1**₅. In the same way, the other proton signals of **1** in G@**1**₅ should be more complicated than those in **1**₆. Such a spectral change was not observed in all the ¹H NMR spectra for the host-guest complexes of the nanocube, which excludes the possibility of the G@**1**₅ open cube.

Figure. (a) A side view of an open nanocube, 1_5 . A red line indicates the sides of the panel where the GSA removed from 1_6 occupied. (b) Views of an open nanocube, 1_5 , from the north and the south poles. $i^1 - i^3$ with an open circle indicate *p*-tolyl methyl groups near the sides that share with the removed GSA (red lines).

REVIEWERS' COMMENTS:

Reviewer #4 (Remarks to the Author):

I have reviewed again the revised version of the manuscript from Hiraoka et al., in which the authors have addressed the issues raised by all referees. I am most grateful to the authors for the new discussions and data, which strengthen the quality of this piece of work.

Unfortunately, my main concern remains: structure elucidation of the encapsulated complexes is weak as it is mainly based on symmetry considerations, changes of spectral data in NMR after encapsulation and in DOSY measurements. I understand that the difficulties in characterizing supramolecular complexes are very high and that the previously mentioned arguments would be enough for publishing in many journals. I also understand that the difficulties in getting X-Ray structures (though these would not confirm structures in solution) and the difficulties for measuring the molecular ion by MS are also very high. But so are the standards of this journal.

Unfortunately, my mind has not changed, even with the new data incorporated: results and their degree of finishing in their current form do not meet the quality expected for publications in Nature Communications. In any case, the final decision on this issue corresponds to the Editor.